# Moth Detection from Pheromone Trap Images Using Deep Learning Object Detectors

**Suk-Ju Hong** [1], **Sang-Yeon Kim** [1], **Eungchan Kim** [1], **Chang-Hyup Lee** [1], **Jung-Sup Lee** [2], **Dong-Soo Lee** [3], **Jiwoong Bang** [2] and **Ghiseok Kim** [1,4,*]

[1] Department of Biosystems and Biomaterials Science and Engineering, Seoul National University, 1 Gwanak-ro, Gwanak-gu, Seoul 08826, Korea; hsj5596@snu.ac.kr (S.-J.H.); yskra@snu.ac.kr (S.-Y.K.); oxycle@snu.ac.kr (E.K.); dlckdguq@snu.ac.kr (C.-H.L.)

[2] Protected Horticulture Research Institute, National Institute of Horticultural and Herbal Science, Rural Development Administration, 1425, Jinham-ro, Haman-myeon, Haman-gun, Gyeongsangnam-do 52054, Korea; suel3841@korea.kr (J.-S.L.); bang21c@korea.kr (J.B.)

[3] Department of Leaders in Industry-University Cooperation, Chung-Ang University, 4726, Seodong-daero, Daedeok-myeon, Anseong-si, Gyeonggi-do 17546, Korea; tara0808@cau.ac.kr

[4] Research Institute of Agriculture and Life Sciences, Seoul National University, 1 Gwanak-ro, Gwanak-gu, Seoul 08826, Korea

[*] Correspondence: ghiseok@snu.ac.kr; Tel.: +82-2-880-4603

**Abstract:** Diverse pheromones and pheromone-based traps, as well as images acquired from insects captured by pheromone-based traps, have been studied and developed to monitor the presence and abundance of pests and to protect plants. The purpose of this study is to construct models that detect three species of pest moths in pheromone trap images using deep learning object detection methods and compare their speed and accuracy. Moth images in pheromone traps were collected for training and evaluation of deep learning detectors. Collected images were then subjected to a labeling process that defines the ground truths of target objects for their box locations and classes. Because there were a few negative objects in the dataset, non-target insects were labeled as *unknown class* and images of non-target insects were added to the dataset. Moreover, data augmentation methods were applied to the training process, and parameters of detectors that were pre-trained with the COCO dataset were used as initial parameter values. Seven detectors—Faster R-CNN ResNet 101, Faster R-CNN ResNet 50, Faster R-CNN Inception v.2, R-FCN ResNet 101, Retinanet ResNet 50, Retinanet Mobile v.2, and SSD Inception v.2 were trained and evaluated. Faster R-CNN ResNet 101 detector exhibited the highest accuracy (mAP as 90.25), and seven different detector types showed different accuracy and speed. Furthermore, when unexpected insects were included in the collected images, a four-class detector with an *unknown class* (non-target insect) showed lower detection error than a three-class detector.

**Keywords:** pheromone trap; pest; moth; deep learning; horticulture; insect detection

---

## 1. Introduction

Continuous monitoring of the pest population is one of the essential elements in the pheromone-based pest control system [1]. Traditional monitoring of pheromone-based traps was conducted by manually identifying the species and population of pests on the trap. This approach is labor intensive and time consuming, and requires skilled personnel capable of distinguishing different species of pests. These disadvantages hinder pest monitoring in real time operation which can guarantee a specified timing constraint. In particular, early diagnosis of pests, one of the major challenges in the horticulture industry, requires automatic monitoring rather than manual monitoring [2].

Recently, research applying IoT (Internet of Things) technologies for farm automation and real-time farm monitoring have been widely conducted [3–5], and the applications of IoT technologies in agriculture enabled real-time remote monitoring of farms using vision systems. Among various vision systems, real-time monitoring equipment using an RGB (Red, Green, and Blue) color camera is fairly configurable because of its relatively inexpensive cost, and it can obtain useful information including shapes, colors and textures. In particular, owing to the rapid development of image processing technology, the effectiveness of RGB image-based monitoring has been increasing in recent years. Therefore, image-based monitoring techniques are being applied in many areas of farm monitoring, such as crop disease monitoring [6,7] and crop growing monitoring [8,9] systems.

Traditional image classification or detection studies generally used image processing methods like active contour [10], scale-invariant feature transform (SIFT) [11], Histogram of Oriented Gradients (HOG) [12], Haar-like features [13] and machine learning methods like support vector machine (SVM) [14], artificial neural network (ANN) [15]. Image-based pest monitoring studies mainly used traditional image processing and machine learning techniques. Xia et al. [16] separated pests and background using watershed segmentation and detected pests by classifying the species of pests using a Mahalanobis distance. Li et al. [17] detected small-sized pests on leaf surfaces by applying multifractal analysis. Wen et al. [18] used a scale-invariant feature transform (SIFT) descriptor and six classifiers, including the minimum least square linear classifier (MLSLC) and K nearest neighbor classifier (KNNC) to classify orchard insects. Wang et al. [19] classified insect images using a support vector machine (SVM) and artificial neural network (ANN) after feature extraction. Bakkay et al. [20] detected moths using adaptive k-means clustering approach by using contour's convex hull and region merging algorithm. Solis-Sanchez et al. [21] applied SIFT descriptor and LOSS algorithm to detect insects on trap. Bechar et al. [22] used pattern recognition methods using local maximality to detect whiteflies.

The detection or classification, using traditional methods, was carried out by a person specifying features manually, and applying the machine learning algorithm after extracting features. However, in order to detect or classify images of various environments, it was necessary to combine not only simple features but also various features that humans cannot understand. Traditional methods of feature extraction and machine learning have made it impossible to create and apply these complex features. Recently, deep-learning methods, based on convolutional neural networks (CNNs), have mainly been used in the field of image classification and detection. In the case of CNN, the biggest difference from traditional methods is that it finds the characteristics and suitable features of target object for a given problem by learning data by itself. If the limitation of the existing methods is to use the features set by humans, CNN can learn and apply complexly from simple features to complex features that are difficult for humans to understand through deep layers.

R-CNN, the first to apply CNN for object detection, outperformed existing object detection models based on feature extraction and machine learning [23]. Faster R-CNN, developed from R-CNN, increases the detection speed by applying a region proposal network (RPN), enabling real-time deep-learning-based detection. Faster R-CNN is 250 times faster than R-CNN and 10 times faster than Fast R-CNN [24], with higher accuracy [25]. After two-stage R-CNN type object detectors, one-stage object detectors, such as you only look once (YOLO) [26], single shot multibox detectors (SSD) [27], and Retinanet [28] were developed. These one-stage detectors showed significantly faster speed than two-stage detectors. Among them, Retinanet showed a high accuracy similar to the two-stage detector and the accuracy of small object detection, which was one of the weak points of the one-stage detector, was also superior to YOLO and SSD.

Deep learning-based object detectors are widely used in vision fields such as pedestrian detection [29], crop disease detection [30], and object detection in aerial images [31,32]. In addition, it is commonly known that detectors based on these deep learning-based detection methods overwhelms the performance of traditional detection methods in most fields of detection application without the traditional labor-intensive feature engineering process. Therefore, in the field of pest detection

from pheromone trap images, studies applying deep-learning methods are increasingly required to accurately detect and classify pests having various characteristics for various environments. In recent years, deep learning-based methods have been applied to pest detection in pheromone trap images. Ding and Taylor [33] detected moths using a sliding window method and CNN. Sun et al. [34] detected a red turpentine beetle (RTB) using a deep learning-based one-stage object detector. Although research has been conducted on pest detection methods in pheromone traps for real-time pest monitoring, it is necessary to study the optimal method considering various deep learning-based object detection methods. The purpose of this study is to develop detectors for three moth species from pheromone trap images using deep learning-based object detection method. For this, pheromone trap images were collected and seven different deep learning-based object detectors were applied to compare their speed and accuracy. The collected data were divided into three sets: Training set, Validation set, and Test set for training and evaluation. Seven different detectors that combine meta architectures such as Faster R-CNN, R-FCN, and SSD and feature extraction networks such as Resnet, Inception, and Mobilenet were trained and compared performance results. In training process, data augmentation and transfer learning were applied.

## 2. Materials and Methods

### 2.1. Data Collection

Figure 1 shows a schematic of the training and evaluation process of this study. As shown in Figure 2, RGB images of moths in pheromone traps were weekly collected during three months period from two local farms located in the Protected Horticulture Research Institute operated by National Institute of Horticultural and Herbal Science in the Republic of Korea. The trap was made of plastic and a pheromone lure was installed in the trap, and glue was treated on trap surface to capture target moth. All images included three types of moths (*Helicoverpa assulta, Spodoptera litura, and Spodoptera exigua*), and they were used during both the learning and evaluation processes of deep learning-based moth detectors. To acquire images under various conditions with a limited number of traps, images were collected using a color camera (acA2440-75uc, Basler Inc., Archbold, OH, USA) that has a 2448 × 2048 pixel resolution, and a 30 degree of horizontal field of view. In addition, two different adjusting illuminations (natural light, white LED light), and two photographing angles (normal directional image, oblique images) were used during the image collection. In addition to acquired moth images from the pheromone traps, we also added various target moth images that are not collected from pheromone traps in order to construct a robust model, as shown in Figure 3. Figure 3 shows the moth images of *Spodoptera litura, Helicoverpa assulta, and Spodoptera exigua*, respectively. Each moth can be distinguished by the pattern of its wings. *Spodoptera litura* moth has brown, complex forewing patterns, *Helicoverpa assulta* moth has fewer patterns than the other two species and *Spodoptera exigua* moth has dot patterns on forewing.

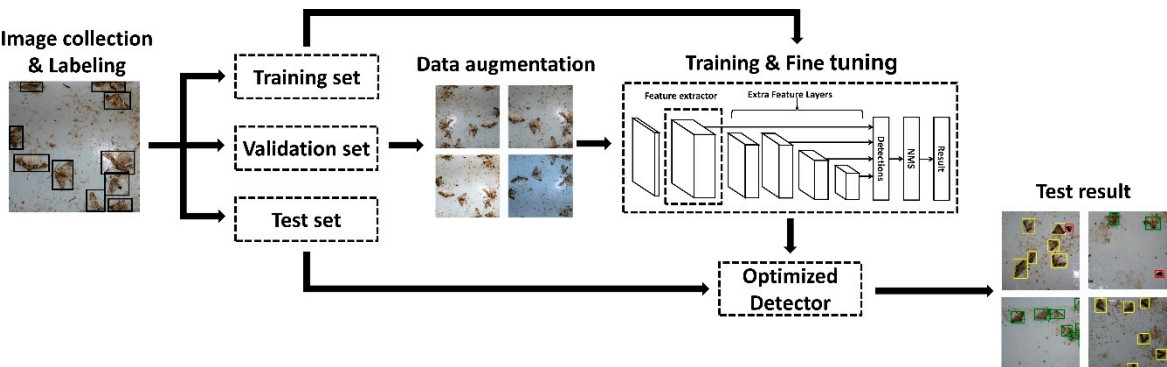

**Figure 1.** Schematic flowchart of the proposed deep-learning process.

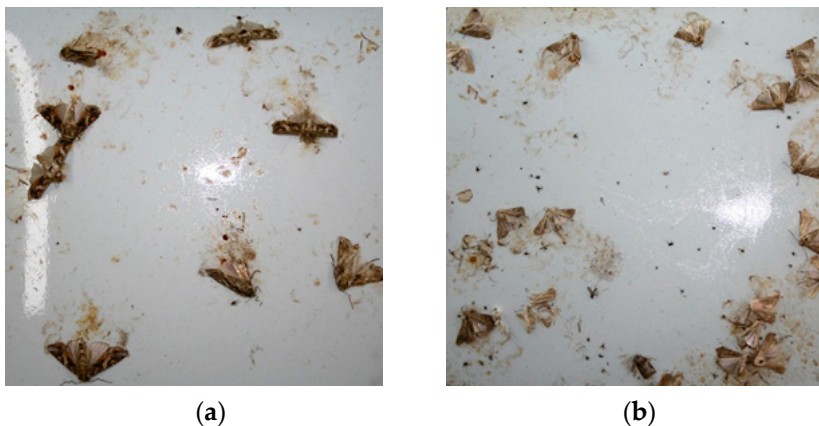

**Figure 2.** Moth images acquired from pheromone trap: (**a**) *Spodoptera litura*; (**b**) *Spodoptera exigua*.

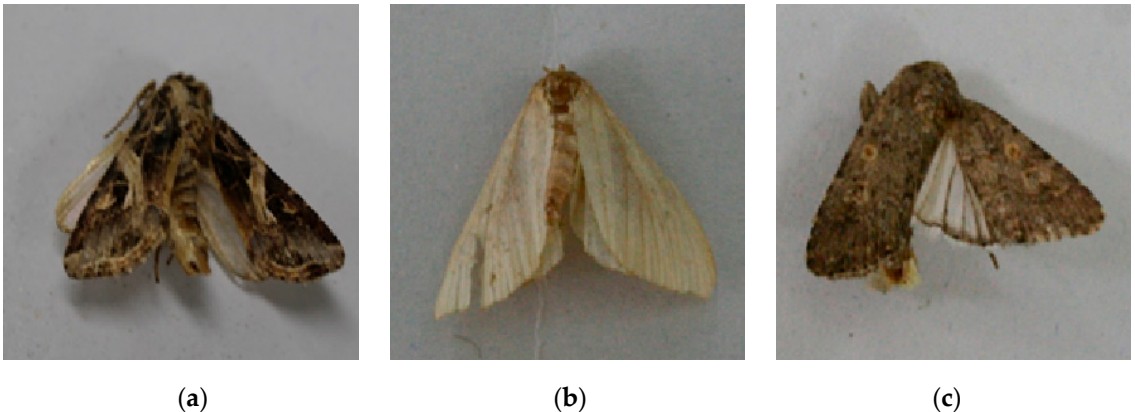

**Figure 3.** Moth images acquired from open dataset: (**a**) *Spodoptera litura*; (**b**) *Helicoverpa assulta*; (**c**) *Spodoptera exigua*.

Pheromone trap images have a simpler background than ordinary images and do not contain various objects. Therefore, there is a lack of difficult negative samples when training and there is a high possibility of false detection when an object not included in the training exists in the image. In this study, insects other than the target moths in the trap image were classified as an *unknown class*. As shown in Figure 4, 168 photos containing various insects were added to the dataset so that non-target insects could be detected as *unknown class* when they were included in the collected image.

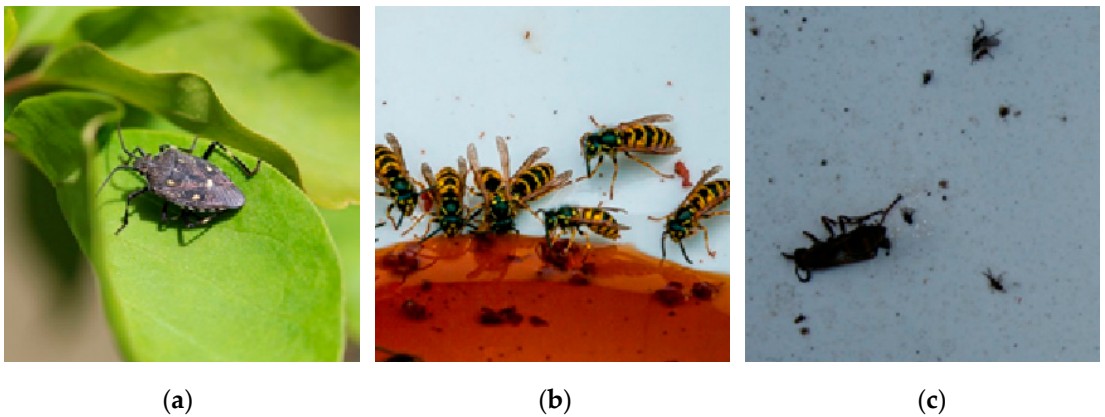

**Figure 4.** Insect images used for the *unknown class* during training: (**a**) Stinkbug; (**b**) Bee; (**c**) Cockroach.

The collected moth images then went through a bounding box labeling process that defines the ground truth boxes and classes of objects in the images. Insects in trap images were labeled with four classes, including three moth classes and an *unknown class* of non-target insects. As a result of data collection and labeling, a total of 1142 images were obtained, 1241 *Spodoptera litura* moths, 621 *Helicoverpa assulta* moths, 1369 *Spodoptera exigua* moth and 986 other insects were labeled.

## 2.2. Detector Training and Evaluation

Training numerous parameters of deep neural networks requires a large amount and diversity of training data. For this reason, data augmentation methods are used to increase the amount and diversity of data by artificially modulating training images. In this study, we employed data augmentation methods to obtain more robust detectors during the training process. Horizontal and vertical flipping, contrast and brightness adjustment, color distortion, random cropping, and 90º rotation were applied to the training process. Each augmentation method was applied with a 50% probability when the data repeatedly entered the detectors. The collected image dataset was divided into train set for training, validation set for detector optimization, and test set for performance evaluation, as shown in Table 1.

**Table 1.** Number of images and insects in images used in training, validation, and testing.

|  | Training | Validation | Testing |
|---|---|---|---|
| Images | 714 | 211 | 217 |
| *Spodoptera litura* moth | 838 | 277 | 254 |
| *Helicoverpa assulta* moth | 356 | 137 | 128 |
| *Spodoptera exigua* moth | 772 | 252 | 199 |
| Other insects | 687 | 154 | 145 |

CNN-based object detection models consist of a combination of meta-architectures such as Faster R-CNN, R-FCN, SSD, and feature extractors such as Alexnet, Mobilenet, Inception, and ResNet. As the combination of meta-architecture and feature extractor changes, the speed and accuracy of the model differ [35]. Therefore, a comparison of these combinations is necessary to select the optimal model. In this study, we trained pest detection models by combining meta-architectures and feature extractors and compared the speed and accuracy of each model. Seven models of Faster R-CNN ResNet 101, Faster R-CNN Inception v.2, Faster R-CNN ResNet 50, R-FCN ResNet 101, Retinanet ResNet 50, Retinanet Mobilenet v.2, and SSD Inception v.2 were trained, compared, and evaluated. Retinanet is named because it uses the ResNet structure. However, in this study, the SSD structure, using focal loss and a feature pyramid network (FPN), is collectively called Retinanet. All detectors were trained after transferring the parameters from the pre-trained detectors with the COCO dataset [36]. Python was used as the programming language, and the training and evaluation code was constructed based on the TensorFlow object detection API [35]. The training and evaluation process were carried out using a computing system with a GeForce GTX 1080ti (Nvidia Corp., Santa Clara, CA, USA) GPU and an Intel core i7-7700k (Intel Corp., Santa Clara, CA, USA) CPU.

The average precision (AP) index was used for detector evaluation. Average precision is the average of precision corresponding to recall from 0 to 1, which is equal to the area under the precision-recall curve. The equations for precision and recall are as shown in Equations (1) and (2), where *TP* is the true positive, that is, the number of moths correctly detected, and *FP* is the false positive, that is, the number of incorrect detections, and *FN* is the false negative, that is, the number of ground truth moths undetected. The correctness of the detection results depends on the threshold of the ground truth box and detection box intersection over union (IOU). In our study, the average precision was calculated based on an IOU threshold of 0.5:

$$\text{Precision} = \frac{TP}{TP + FP} \tag{1}$$

$$\text{Recall} = \frac{TP}{TP + FN} \tag{2}$$

## 3. Results and Discussion

Figure 5 shows the precision-recall (PR) curve of the trained detectors. The PR curve of *Spodoptera litura* shown in Figure 5a,d maintained high precision, close to 1.0 in all seven types of detectors, even in the high recall area. However, in the PR curve of *Helicoverpa assulta* in Figure 5b,e, as the Recall increased, the Precision was significantly reduced compared to the PR curve results of the other two types of moths, especially for the SSD Inception v.2 model. In the PR curve of *Spodoptera exigua* in Figure 5c,f, the high precision was maintained in all recall areas, except for the SSD Inception v.2 detector. Figure 6 and Table 2 show the detector's average precision and processing time of detectors per image, i.e., the inference time, which can be defined as the processing time of taking a model that has already been trained and using that trained model to make a useful prediction. The Faster R-CNN ResNet 101 detector had the longest inference time of 103 ms and the highest mAP of 90.25. In comparison with Faster R-CNN ResNet 101, the faster R-CNN Inception v.2 detector showed similar accuracy (mAP 90.05) despite the shorter inference time of 72 ms. Faster R-CNN ResNet 50 showed a high mAP of 88.62, but it had a longer inference time and lower mAP than the Faster Inception v.2 model. The R-FCN ResNet101 detector had a medium speed and accuracy among the detectors with 86.91 mAP and 67 ms inference time. The Retinanet ResNet 50 detector, which is a one-stage detector, showed high accuracy (mAP 88.62) similar to that of two-stage detectors. The Retinanet Mobilenet v.2 detector showed high accuracy (mAP 85.21) despite its high speed of 34 ms inference time. The SSD Inception v.2 detector showed the shortest inference time of 23 ms, but a much lower accuracy (mAP 76.86) than the other models. In total, test results showed a detector's trade-off relationship between the inference time and the mAP.

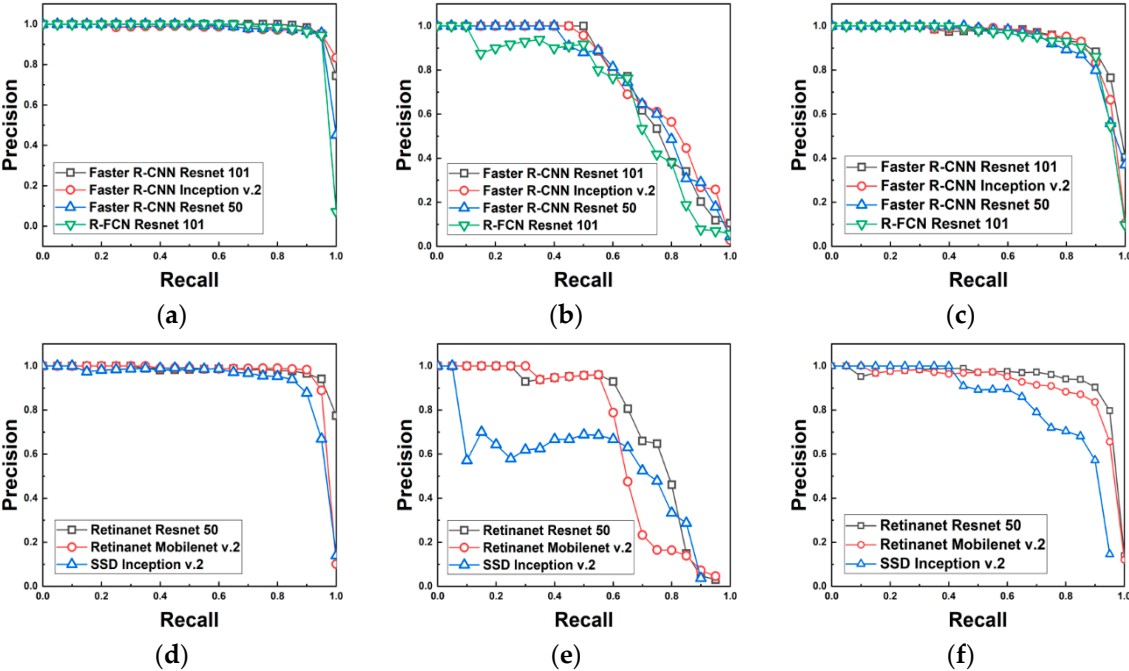

**Figure 5.** Precision-Recall(PR) curve of trained detectors: (**a**) Faster R-CNN based models and R-FCN model for *Spodoptera litura* moth; (**b**) Faster R-CNN based models and R-FCN model for *Helicoverpa assulta* moth; (**c**) Faster R-CNN based models and R-FCN model for *Spodoptera exigua* moth; (**d**) SSD-based models for *Spodoptera litura* moth; (**e**) SSD-based models for *Helicoverpa assulta* moth; (**f**) SSD-based models for *Spodoptera exigua* moth.

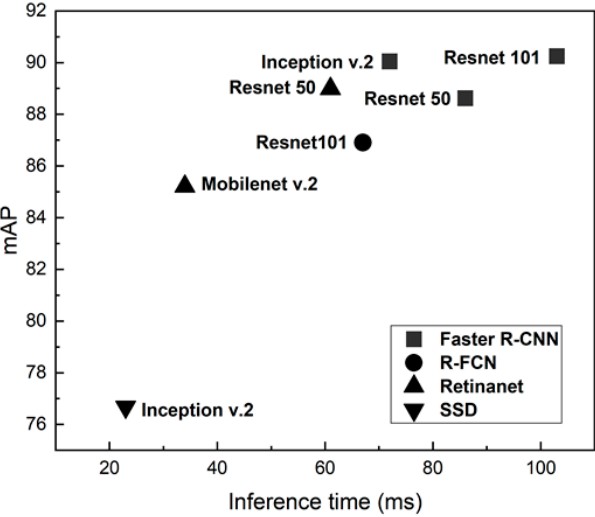

**Figure 6.** Mean average precision (mAP)-Inference time plot for trained detectors.

**Table 2.** Test results of moth detection models.

| Meta Architecture | Feature Extractor | Inference Time (ms/image) | mAP | AP(SL) * | AP(HA) * | AP(SE) * |
|---|---|---|---|---|---|---|
| Faster R-CNN | ResNet 101 | 103 | 90.25 | 98.06 | 77.59 | 95.11 |
| | Inception v.2 | 72 | 90.05 | 98.43 | 77.35 | 94.37 |
| | ResNet 50 | 95 | 88.62 | 98.50 | 75.28 | 92.09 |
| R-FCN | ResNet 101 | 67 | 86.91 | 98.02 | 70.21 | 92.51 |
| Retinanet | ResNet 50 | 61 | 88.99 | 98.28 | 74.13 | 94.56 |
| | Mobilenet v.2 | 34 | 85.21 | 97.76 | 66.53 | 91.34 |
| SSD | Inception v.2 | 23 | 76.86 | 93.41 | 55.19 | 81.97 |

* AP(SL) is the average precision of *Spodoptera litura*, AP(HA) is the average precision of *Helicoverpa assulta*, and AP(SE) is the average precision of *Spodoptera exigua*.

Figure 7 shows the representative moth detection results using the Faster R-CNN ResNet 101 detector, which shows the highest mAP value among all the models in this study. As shown in Figure 7, all the moth and other insects (*unknown class*) were successfully detected and three types of moth classes (*Spodoptera litura, Helicoverpa assulta, and Spodoptera exigua*) and *unknown class* were marked with yellow, white, green, and red boxes, respectively. Figure 8 shows the erroneous detection results. As shown in Figure 8a, there are some errors in which several moths are detected as one object when the moths are overlapped or held adjacently. Figure 8b is the most common error in which the wings of an unrecognizable moth falling in a trap are incorrectly detected as *Helicoverpa assulta*. Figure 8c shows the error of confusion with other moth classes when the wing pattern is not revealed on the image because the body of the moth faces the front not a wing. In particular, the detection errors were the most frequent in *Helicoverpa assulta*. Comparing the detector's average precision by class, the APs of *Spodoptera litura* and *Spodoptera exigua* ranged from 93.41 to 98.06, and from 81.97 to 95.11, respectively, while APs of *Helicoverpa assulta* ranged from 55.19 to 77.59, which were fairly lower than the others. We assumed that the detection errors shown in Figure 8 could be caused by several complex reasons. The first reason is the relatively small number of *Helicoverpa assulta* image datasets, so the various data was not learned during the training process. The second reason is the weak wing pattern of *Helicoverpa assulta* among the three moth classes used in this study. Commonly, when the forewings of moths are not exposed in the collected image, only their hind wings are identified. In this case of unclear wing patterns being tested, the moths are sometimes mistakenly considered as *Helicoverpa assulta*. Additionally, when the torn wings are attached to the traps, such as in Figure 8b, often wing patterns are not revealed, in which case they are often recognized as *Helicoverpa assulta*. In addition, error cases in which wings of other insects that had no pattern were sometimes recognized as *Helicoverpa assulta*.

We expect that these incorrect situations can be improved when images of the *Helicoverpa assulta* moth, under various states and torn wing data, are sufficiently added to the train dataset so that *Helicoverpa assulta* detection accuracy increases and the torn wings are not recognized as moths.

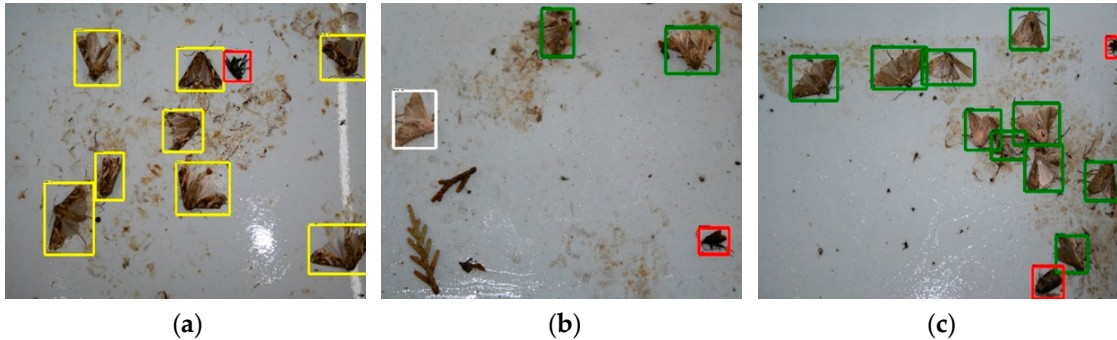

**Figure 7.** Moth detection results of Faster R-CNN ResNet 101 detector with a confidence score threshold of 50%: (**a**) *Spodoptera litura* (yellow box) and unknown insect (red box); (**b**) *Helicoverpa assulta* (white box) and unknown insect (red box); (**c**) *Spodoptera exigua* (green box) and unknown insects (red box).

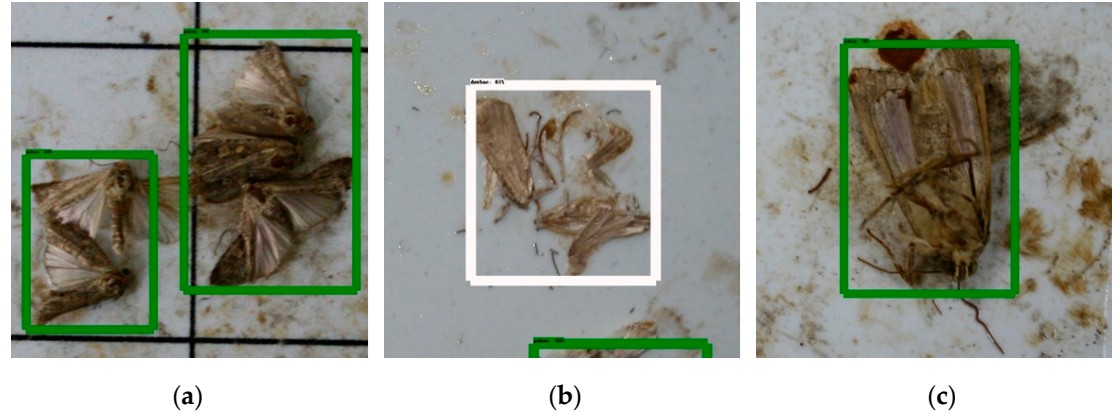

**Figure 8.** Erroneous detection results with a confidence score threshold of 50%: (**a**) Overlapped moths were detected as single moth; (**b**) Moth with torn wings was detected as *Helicoverpa assulta*; (**c**) Classification failure caused by an unrevealed wing pattern on the image.

Table 3 shows the comparison results of the detector with and without the images of the *unknown class*. For comparison of average precision in the same test set, a test set was constructed with only target moth data images excluding *unknown class* images. Although new class (*unknown class*) was added on the model and only moth trap images were tested, there was improvement in mean average precision. Figure 9 shows the detection result images of the three-class and four-class detectors. Comparing Figure 9a,b, which are the result images of the three-class and four-class detectors, respectively, the fly that was confused with the moth in the three-class detector was correctly classified as an *unknown class* in the four-class detector. In addition, the leaf that was incorrectly detected as a moth in the three-class detector was not detected as a moth in the four-class detector. The moth trap image set used for the comparison of the three and four class models did not contain many other insects or objects, so there was no significant difference in the average precision value. However, when data other than moth trap data was input, the error of the three class detector increased as shown in Figure 9c. In Figure 9c, the three-class detector incorrectly detected wasps as the moth. However, in Figure 9d, the result of the four-class detector, wasps were detected as *unknown class* without being confused with the moth. As shown in Table 3, the average precision of the Faster R-CNN ResNet 101 detectors for three target moth species with and without the image of an *unknown class* (three-class and four-class detectors) was not significantly improved. However, the four-class model

with the images of *unknown class* successfully detected non-moth objects and showed stable detection results when objects other than moths were included in the image shown in Figure 9.

**Table 3.** Test results with and without unknown class images.

| Number of Class | Meta Architecture | Feature Extractor | mAP | AP(SL) * | AP(HA) * | AP(SE) * |
|---|---|---|---|---|---|---|
| 3 | Faster R-CNN | ResNet 101 | 89.64 | 98.08 | 76.65 | 94.21 |
| 4 | Faster R-CNN | ResNet 101 | 90.25 | 98.06 | 77.59 | 95.11 |

* AP(SL) is the average precision of *Spodoptera litura*, AP(HA) is the average precision of *Helicoverpa assulta*, and AP(SE) is the average precision of *Spodoptera exigua*.

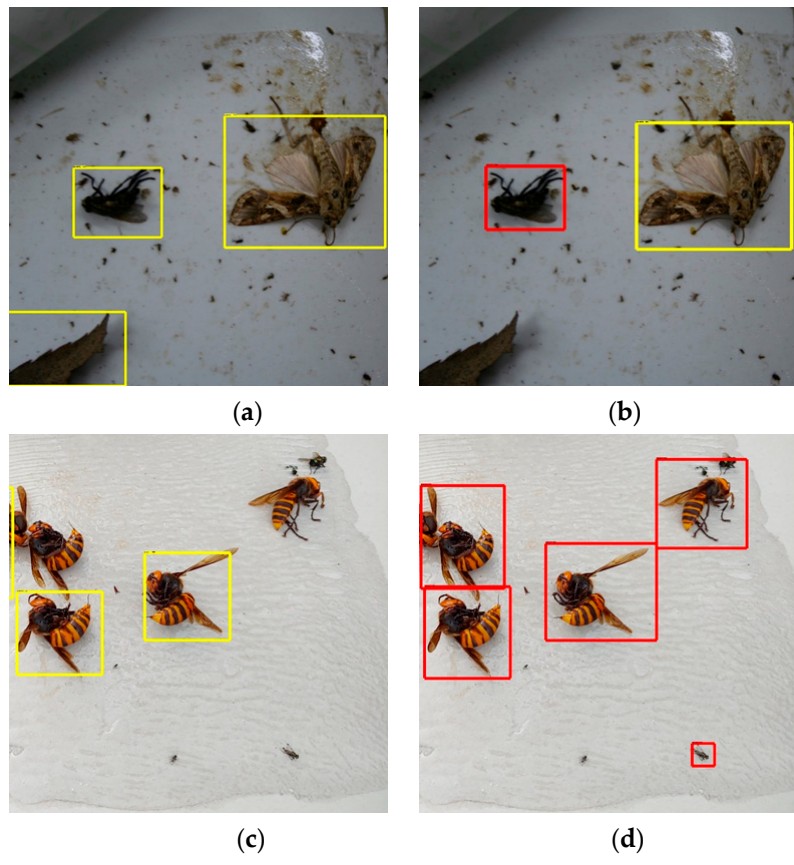

**(a)** **(b)**

**(c)** **(d)**

**Figure 9.** Detection result images of the three-class detector; (**a**,**c**) and four-class detector (**b**,**d**).

## 4. Conclusions

In this study, seven different deep learning object detectors were applied to create detection models for three species of moths collected from pheromone traps. The speed and accuracy of the seven detectors were compared to enable the selection of a suitable model according to the performance and purpose of the pest control system. These results showed that deep-learning-based object detectors are fairly applicable to pheromone-based pest control systems.

Although we suggested deep learning-based moth detection models, with high performance of mAP 90.25, through a training process with image datasets consisting of both, moth images from trap and non-target insect images, higher performance will be expected if more moth images from traps and diverse environments are collected and used in the training process. If a large pest image dataset that includes more diverse species of insects from the trap is constructed, a more precise and robust detector can be achieved without the addition of non-target insect images. In particular, in this study there were relatively fewer images of the *Helicoverpa assulta* moth than those of the other two species. Therefore, a more enhanced detection performance is expected when the image dataset of

*Helicoverpa assulta* moth is supplemented. However, the collection of specific moth images, such as *Helicoverpa assulta* takes a lot of time and effort because it cannot be easily controlled by humans.

Moreover, the image dataset, used in this study, includes some limitations because they are not real-time remote sensing images from the insect trap in the farm field. The moth images, used in this study, were prepared by taking the insect collection area, such as the glue board, after it was manually collected from each trap. However, in the actual moth images from the trap in a real farm field, there will be some undesirable light interference, such as reflection by illumination of natural or artificial light sources. Therefore, in order to create a real-time moth detection system that operates more robustly in a real farm field, a deep learning-based moth detection model needs to be updated using the real-time collected moth images, which include various environmental light effects. For this purpose, it is essential to develop a smart trap system which has a small sized high performance Wi-Fi camera and optimized light source to collect target object images and to analyze the types and population of pests caught by the trap. These works will enable the standardization of the acquisition process of target images from smart traps.

Despite some limitations in this study, we observed that the developed deep learning-based moth detection models showed a mAP of 90.25 for the detection of three species of pest moths in horticulture farms. In our next study, we are developing more robust and precise moth detection models through image collection from in-field traps and optimization of deep learning object detectors by modifying the network structure of possible architectures.

**Author Contributions:** S.-J.H. designed the experimental concept and data analysis methods; S.-Y.K. and E.K. performed experiments and analyzed the data; C.-H.L. supported experiments and data analysis; J.-S.L. supported data analysis; D.-S.L. and J.B. collected and preprocessed data; G.K. wrote the article and have been supervising, discussing the experiments and edited the presented work. All authors have read and agreed to the published version of the manuscript.

**Funding:** This work was supported by Korea Institute of Planning and Evaluation for Technology in Food, Agriculture and Forestry(IPET) through Research and Development Program for Smart Farm(Plant) Project, funded by Ministry of Agriculture, Food and Rural Affairs(MAFRA)(319007-01-1-HD020).

**Conflicts of Interest:** The authors declare no conflict of interest.

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
