# Peer review of "Moth Detection from Pheromone Trap Images Using Deep Learning Object Detectors"

_agriculture, doi:10.3390/agriculture10050170_

Round 1

Reviewer 1 Report

The paper is well written and most of the explanations are clear. The main concern of the paper is that the novelty is not enough justified and highligthted. Moreover, the state of the art description can be improved. Most of my comments are for improving these aspects.

In the introduction, in order to better highlight your contribution and the interest, I suggest to :
1) Develop more the traditional approaches in computer vision : pattern recognition, segmentation, points of interest -> what are the limits ?
2) Develop more traditional approaches in pest moth recognition -> what are the limits ?
3) Develop the possible neuronal network and why CNN is sufficient -> Why choosing network ?
4) At the end of section 1 it should be clear that it is needed to work with deep learning and what the proposed approach brings in comparison to the other exsiting approaches.
5) At the end of section 1, it is nice to introduce the outline.

The description of the state of the art is globally nice but I think that some references are missing, in particular outside the specific scope of pest moth, like :
- For classical computer vision techniques :
-- Active contours : Kass, M., Witkin, A., Terzopoulos, D.: ‘Snakes: active contour models’, Int. J.Comput. Vis., 1988, 1, (4), pp. 321–331
-- Classical segmentation approach : Comaniciu, D., Meer, P.: ‘Mean shift: a robust approach toward feature spaceanalysis’, IEEE Trans. Pattern Anal. Mach. Intell., 2002, 24, (5), pp. 603–619
-- SIFT, that is cited in the paper : Lowe, D.: ‘Distinctive image features from scale-invariant keypoints’, Int. J.Comput. Vis., 2004, 60, (2), pp. 91–110
- For other insects, but similar problem : M. C. Bakkay, S. Chambon, H. A. Rashwan, C. Lubat and S. Barsotti, "Automatic detection of individual and touching moths from trap images by combining contour-based and region-based segmentation," in IET Computer Vision, vol. 12, no. 2, pp. 138-145, 3 2018.
I think that these ICPR workshop can be a good start for enriching the bibliography : http://homepages.inf.ed.ac.uk/rbf/vaib20.html
- For a state of the art on deep learning : I. Goodfellow, Y. Bengio et A. Courville. Deep learning. MIT Press, 2016. http://www.deeplearningbook.org.

About the input data, I have many questions :
- What are the difficulties ? I can see on figure 1 or 3.c, some problems : the moths are deteriorated, the background can be noisy ... I wonder : where is the pheromon ? Is there glue on the support ? All the aspects are problems and difficulties so it has to be explained, to be discussed. This is what is interesting in this research.
- How the data are really labelled ? Bounding boxes ? Positions ? Who labelled them ? Researchers ? Experts ? It is a minor question but it is still interesting.

About the method : having a schema and even a resume of the network used in a figure can be helpful. Moreover, what is the interest of testing all these networks ? Can you justify the testing of all these configurations ?

In the conclusion, the authors suggest that more training data are necessary but I can also suggest adversarial generative network and more specifically autoencoder : Bengio, Y. (2009). Learning deep architectures for AI. Foundations and trends® in Machine Learning, 2(1), 1-127.

Author Response

Responses to the Reviewer #1’s comments

Dear Reviewers

We appreciate your precious comments that help to improve the clearness and exactness of our manuscript. Following your comments, we have corrected our manuscript as follows:

Responses to Reviewer's comments,

The paper is well written and most of the explanations are clear. The main concern of the paper is that the novelty is not enough justified and highligthted. Moreover, the state of the art description can be improved. Most of my comments are for improving these aspects.

In the introduction, in order to better highlight your contribution and the interest, I suggest to :

Comment or Suggestions 1)

Develop more the traditional approaches in computer vision : pattern recognition, segmentation, points of interest à what are the limits ?

Answer or Correction 1)

We added more references about traditional approaches in computer vision at line 54-57. And, some limitations of traditional approaches were added at line 69-74 as follows.

Line 54-57) “Traditional image classification or detection studies generally used image processing methods like active contour [10], scale-invariant feature transform (SIFT) [11], Histogram of Oriented Gradients (HOG) [12], Haar-like features [13] and machine learning methods like support vector machine (SVM) [14], artificial neural network (ANN) [15].”

Line 69-74) “In the case of detection or classification using traditional methods, it was done by a person through specifying features manually, and applying the machine learning algorithm after extracting features. However, in order to detect or classify images of various environments, it was necessary to combine not only simple features but also various features that humans cannot understand. Traditional methods of feature extraction and machine learning have made it impossible to create and apply these complex features.”

Comment or Suggestions 2)

Develop more traditional approaches in pest moth recognition à what are the limits ?

Answer or Correction 2)

We added more references of traditional approaches in pest moth recognition at line 65-68 as follows.

Line 65-68) “Bakkay et al. [20] detected moths using adaptive k-means clustering approach by using contour’s convex hull and region merging algorithm. Solis-Sanchez et al. [21] applied SIFT descriptor and LOSS algorithm to detect insects on trap. Bechar et al. [22] used pattern recognition methods using local maximality to detect whiteflies.”

We believe that the limitations of traditional approaches described at line 69-74 can be considered as the limitations of traditional approaches in pest moth recognition.

Comment or Suggestions 3)

Develop the possible neuronal network and why CNN is sufficient à Why choosing network ?

Answer or Correction 3)

We added some sentence to describe the benefit of CNN at line 75-79 as follows.

Line 75-79) “In the case of CNN, the biggest difference from traditional methods is that it finds the characteristics and suitable features of target object for a given problem by learning data by itself. If the limitation of the existing methods is to use the features set by humans, CNN can learn and apply complexly from simple features to complex features that are difficult for humans to understand through deep layers.”

Comment or Suggestions 4)

At the end of section 1 it should be clear that it is needed to work with deep learning and what the proposed approach brings in comparison to the other existing approaches.

Answer or Correction 4)

We added the why deep learning is needed and what the proposed approach brings in comparison to the other existing approaches at line 91-97 as follows.

Line 91-97) “In addition, it is commonly known that detectors based on these deep learning-based detection methods overwhelms the performance of traditional detection methods in most fields of detection application without the traditional labor-intensive feature engineering process. Therefore, in the field of pest detection from pheromone trap images, studies applying deep learning methods are increasingly required to accurately detect and classify pests having various characteristics for various environments.”

Comment or Suggestions 5)

At the end of section 1, it is nice to introduce the outline.

Answer or Correction 5)

Reflecting your suggestion, we revised and added line 102-109 as follows.

Line 102-109) “The purpose of this study is to develop detectors for three moth species from pheromone trap images using deep learning-based object detection method. For this, pheromone trap images were collected and seven different deep learning-based object detectors were applied to compare their speed and accuracy. The collected data were divided into three sets: Training set, Validation set, and Test set for training and evaluation. Seven different detectors that combine meta architectures such as Faster R-CNN, R-FCN, and SSD and feature extraction networks such as Resnet, Inception, and Mobilenet were trained and compared performance results. In training process, data augmentation and transfer learning were applied.

Comment or Suggestions 6)

The description of the state of the art is globally nice but I think that some references are missing, in particular outside the specific scope of pest moth, like :

- For classical computer vision techniques :

-- Active contours : Kass, M., Witkin, A., Terzopoulos, D.: ‘Snakes: active contour models’, Int. J.Comput. Vis., 1988, 1, (4), pp. 321?331

-- Classical segmentation approach : Comaniciu, D., Meer, P.: ‘Mean shift: a robust approach toward feature spaceanalysis’, IEEE Trans. Pattern Anal. Mach. Intell., 2002, 24, (5), pp. 603?619

-- SIFT, that is cited in the paper : Lowe, D.: ‘Distinctive image features from scale-invariant keypoints’, Int. J.Comput. Vis., 2004, 60, (2), pp. 91?110

- For other insects, but similar problem : M. C. Bakkay, S. Chambon, H. A. Rashwan, C. Lubat and S. Barsotti, "Automatic detection of individual and touching moths from trap images by combining contour-based and region-based segmentation," in IET Computer Vision, vol. 12, no. 2, pp. 138-145, 3 2018.

I think that these ICPR workshop can be a good start for enriching the bibliography : http://homepages.inf.ed.ac.uk/rbf/vaib20.html

- For a state of the art on deep learning : I. Goodfellow, Y. Bengio et A. Courville. Deep learning. MIT Press, 2016. http://www.deeplearningbook.org.

Answer or Correction 6)

First, thanks for your kind description about helpful references. For your suggestion, we added nine references at line 54-57 and 65-68 to describe classical machine vision techniques and similar insect detection papers as follows.

  1. Kass, M.; Witkin, A.; Terzopoulos, D. Snakes: Active contour models. Int. J. Comput. Vis. 1988, 1, 321–331.
  2. Lowe, D.G. Distinctive image features from scale-invariant keypoints. Int. J. Comput. Vis. 2004, 60, 91–110.
  3. Dalal, N.; Triggs, B. Histograms of Oriented Gradients for Human Detection.; IEEE, 2005; Vol. 1, pp. 886–893.
  4. Viola, P.; Jones, M. Rapid object detection using a boosted cascade of simple features. In Proceedings of the Proceedings of the IEEE Computer Society Conference on Computer Vision and Pattern Recognition; 2001; Vol. 1.
  5. Suykens, J.A.K.; Vandewalle, J. Least squares support vector machine classifiers. Neural Process. Lett. 1999, 9, 293–300.
  6. McCulloch, W.S.; Pitts, W. A logical calculus of the ideas immanent in nervous activity. Bull. Math. Biophys. 1943, 5, 115–133.
  7. Bakkay, M.C.; Chambon, S.; Rashwan, H.A.; Lubat, C.; Barsotti, S. Automatic detection of individual and touching moths from trap images by combining contour-based and region-based segmentation. IET Comput. Vis. 2018, 12, 138–145.
  8. Solis-Sánchez, L.O.; Castañeda-Miranda, R.; García-Escalante, J.J.; Torres-Pacheco, I.; Guevara-González, R.G.; Castañeda-Miranda, C.L.; Alaniz-Lumbreras, P.D. Scale invariant feature approach for insect monitoring. Comput. Electron. Agric. 2011, 75, 92–99.
  9. Bechar, I.; Moisan, S.; Thonnat, M.; Bremond, F. On-line video recognition and counting of harmful insects. In Proceedings of the Proceedings - International Conference on Pattern Recognition; 2010; pp. 4068–4071.

Comment or Suggestions 7)

About the input data, I have many questions :

- What are the difficulties ? I can see on figure 1 or 3.c, some problems : the moths are deteriorated, the background can be noisy

Answer or Correction 7)

As you commented, deteriorated or overlapped moths shown in figure 2 or 4c (figure number was updated) may be acted as the cause of the most error (figure 8), and we described the situations and possible solutions for improvement at line 222-227 and line 230-242. However, we think that the noisy background shown in figure 2 did not seriously affect the detection performance of developed models. For such a reason, we collected moth images under various conditions to prevent those noisy background be a trouble reason, and we added below sentence at line 122-124 as follows.

Line 122-124) “In addition, two different adjusting illuminations (natural light, white LED light), and two photographing angles (normal directional image, oblique images) were used during the image collection.”

Comment or Suggestions 8) I wonder : where is the pheromone? Is there glue on the support? All the aspects are problems and difficulties so it has to be explained, to be discussed. This is what is interesting in this research.

Answer or Correction 8)

Pheromone lure was installed in sticky trap, and glue was treated on trap surface to capture moths. We added that description at line 117 as follows.

Line 117) “The trap was made of plastic and a pheromone lure was installed in the trap, and glue was treated on trap surface to capture target moth.”

Comment or Suggestions 9) How the data are really labelled? Bounding boxes? Positions?

Comment or Suggestions 9)

We used bounding box labelling in this study. To clarify this, line 148 was revised as follows.

“The collected moth images then went through a bounding box labeling process that defines the ground truth boxes and classes of objects in the images.”

Comment or Suggestions 10)

Who labelled them? Researchers? Experts? It is a minor question but it is still interesting.

Comment or Suggestions 10)

The labeling was conducted by an insect expert, after being trained by researchers.

Comment or Suggestions 11)

About the method : having a schema and even a resume of the network used in a figure can be helpful.

Comment or Suggestions 11)

We added schematic flowchart of the proposed deep-learning process as Figure 1.

Comment or Suggestions 12)

Moreover, what is the interest of testing all these networks? Can you justify the testing of all these configurations?

Comment or Suggestions 12)

The reason for testing and comparing various networks is to compare the speed and accuracy performances of each developed model to select a fast or light and optimal model. In general cases, the slower the speed, the higher the accuracy performance. On the contrary, the higher the speed, the lower the precision performance. However, it may appear differently depending on the network used and the applied data set. Therefore, we should perform test process for each developed model.

Comment or Suggestions 13)

In the conclusion, the authors suggest that more training data are necessary but I can also suggest adversarial generative network and more specifically autoencoder : Bengio, Y. (2009). Learning deep architectures for AI. Foundations and trends® in Machine Learning, 2(1), 1-127.

Comment or Suggestions 13)

Thanks you so much this information. I didn’t know well about it. As we wrote in the conclusion, we are now collecting more moth images from different traps. This work takes a lot of time and labor intensive. Therefore, your information will be very helpful to us. I will positively consider your suggestion for our on-going studies. Thanks you.

Reviewer 2 Report

This manuscript is about a comparison of using different deep-learning image analysis networks to detect and identify a number of species of insects (mainly moths) on pheromone sticky traps. This topic is very relevant for advancing the further automation of registration systems in horticulture and agriculture, to reduce manual labor and to provide data to decision support systems in the future. Quite a number of research groups but also companies in the field of biological pest control are currently working on similar topics and applications.

The manuscript reads well is easy to follow and is understandable. However, especially some conclusions drawn are done with soft and subjective expressions like: it worked good, robust. This should be changed and more objective and measurable criteria should be used instead.

Abstract: “Diverse pheromones and pheromone-based traps have been studied” is not precise and should be better changed to something like:  “Images acquired from insects captured by pheromone-based traps have been studied”

Keywords: Keywords like moth and pheromone trap are already part of the title and for most journals there is no need to add title words as keywords again. I suggest to add insect or insect detection to the keywords.

I noticed some grammar issues and sometimes the chosen words might be not correct. A check is required.  

General: to my feeling the literature cited in the introduction has a bias towards publications from authors from Asian countries. There is a more published on the topic of pest detection (using image analysis for example) also in other parts of the world recently. I would like to suggest that the authors review again/include other sources also.

In the introduction it would be nice to be a bit more specific on the requirements of such a system. You talk about real-time (L42), what level of accuracy/detection rate would be required for an automatic system? In the conclusion section (L245) talk about the performance and purpose. You could better discuss this in the results & discussion section. The discussion is put together with the results. It would be clearer to have separate sections for both. Also, it would be good to add some discussion how the results of your research compare to results found in literature.

The Materials and Methods section is incomplete and not detailed enough. Images were collected from how many farms? How long have the traps been exposed to the insects, how long have the insects been dead (and possibly dehydrated and therefore changed shape or changed color)? How was the illumination adjusted for image acquisition? What illumination? What camera with what resolution was used?  What was the background material? Without that information it would not be possible to repeat the experiment. In Figure 1 and 2 a white background is visible, in Figure 3 also other colors, did that not influence the results? Later in the conclusion section (L258 ff) you talk a bit about that, please consider to move this part to the material and methods section. If I look to Figure 8 c and d it seems that the other insects are also on white background only. It is also unclear how many extra images/object instances were created by the data augmentation. Are the numbers in table 1 before or after data augmentation? Also, what was the specs of the computer used for analysis (with respect to be able to asses the inference time)?

A comment/question on your method: when the deep-learning network detects an object it also provides a confidence score (0 to 100%) about the detection (typically shown in the upper right corner of the bounding box). It remains unclear if you used all detections or if you applied a threshold to this score in order to count only detections with a high confidence. Please elaborate on this. Using such a threshold may also be helpful to exclude false positives for the three-class detector (as shown in figure 8 c) as most likely such detections have a low confidence value.

More detailed comments:

L33: “more stable detection” more stable than what? The detection has a certain precision and accuracy and results between different methods might be different. But stable is a very subjective word that should be avoided.

L42: Please elaborate why real-time monitoring is needed. I do see that automatic procedures are needed, but why real-time. You have first to define what you understand as real-time.

L47: RGB color sensor. I guess you mean color camera, please be more specific as a color sensor does not necessarily needs to generate 2d images.

L64: “reduces the dectection speed”  is wrong. It increases the detection speed (or reduces the time needed).

L140/L141 reference 27 is mentioned before 26. Numbers should be in ascending numbers according to the appearance in the text.

Table 2: I would rather call it “ms/image”  instead of  “ms/photograph”.

L219: minute improvement? Do you mean: minor improvement?

L246: you talk about “good” performance. However, it remains completely undefined what good or bad is.

L270: also here you say that it “worked well”. This is very subjective and should be avoided. I miss the evaluation against the (to be defined) requirements or how your results compare to other research.

Author Response

Responses to the Reviewer #2’s comments

Dear Reviewers

We appreciate your precious comments that help to improve the clearness and exactness of our manuscript. Following your comments, we have corrected our manuscript as follows:

Responses to Reviewer's comments,

Comments and Suggestions for Authors

This manuscript is about a comparison of using different deep-learning image analysis networks to detect and identify a number of species of insects (mainly moths) on pheromone sticky traps. This topic is very relevant for advancing the further automation of registration systems in horticulture and agriculture, to reduce manual labor and to provide data to decision support systems in the future. Quite a number of research groups but also companies in the field of biological pest control are currently working on similar topics and applications.

The manuscript reads well is easy to follow and is understandable. However, especially some conclusions drawn are done with soft and subjective expressions like: it worked good, robust. This should be changed and more objective and measurable criteria should be used instead.

Comment or Suggestions 1)

Abstract: “Diverse pheromones and pheromone-based traps have been studied” is not precise and should be better changed to something like: “Images acquired from insects captured by pheromone-based traps have been studied”

Answer or Correction 1)

We revised sentences of lines 18-20 to reflect your suggestion as follows.

“Diverse pheromones and pheromone-based traps as well as images acquired from insects captured by pheromone-based traps have been studied and developed to monitor the presence and abundance of pests and to protect plants.”

Comment or Suggestions 2)

Keywords: Keywords like moth and pheromone trap are already part of the title and for most journals there is no need to add title words as keywords again. I suggest to add insect or insect detection to the keywords.

Answer or Correction 2)

We added keyword of “insect detection” instead of object detection at line 35 to reflect your suggestion.

Comment or Suggestions 3)

I noticed some grammar issues and sometimes the chosen words might be not correct. A check is required.

Answer or Correction 3)

Reflecting your suggestions, I have checked whole pages to enhance writings, and the revised manuscript was proofread by native speaker, and we attached a certificate of proofreading.

Comment or Suggestions 4)

General: to my feeling the literature cited in the introduction has a bias towards publications from authors from Asian countries. There is a more published on the topic of pest detection (using image analysis for example) also in other parts of the world recently. I would like to suggest that the authors review again/include other sources also.

Answer or Correction 4)

Reflecting your suggestion, we added non-Asian studies about pest detection as references as follows.

  1. Bakkay, M.C.; Chambon, S.; Rashwan, H.A.; Lubat, C.; Barsotti, S. Automatic detection of individual and touching moths from trap images by combining contour-based and region-based segmentation. IET Comput. Vis. 2018, 12, 138–145.
  2. Solis-Sánchez, L.O.; Castañeda-Miranda, R.; García-Escalante, J.J.; Torres-Pacheco, I.; Guevara-González, R.G.; Castañeda-Miranda, C.L.; Alaniz-Lumbreras, P.D. Scale invariant feature approach for insect monitoring. Comput. Electron. Agric. 2011, 75, 92–99.
  3. Bechar, I.; Moisan, S.; Thonnat, M.; Bremond, F. On-line video recognition and counting of harmful insects. In Proceedings of the Proceedings - International Conference on Pattern Recognition; 2010; pp. 4068–4071.

Comment or Suggestions 5)

In the introduction it would be nice to be a bit more specific on the requirements of such a system. You talk about real-time (L42), what level of accuracy/detection rate would be required for an automatic system?

Answer or Correction 5)

Real time system we mentioned at line 42 means that the system subjected to real time, i.e., response should be guaranteed within a specified timing constraint or system should meet the specified deadline. Of course, the higher a level of accuracy the better. We revised a sentence of line 42 as follows.

“These disadvantages hinder pest monitoring in real time operation which can guarantee a specified timing constraint.”

Comment or Suggestions 6)

In the conclusion section (L245) talk about the performance and purpose. You could better discuss this in the results & discussion section. The discussion is put together with the results. It would be clearer to have separate sections for both. Also, it would be good to add some discussion how the results of your research compare to results found in literature.

Answer or Correction 6)

In the chapter of ‘Results and Discussion’, we described the performance of each detection model by comparing their mean average precisions, and inference times for detection accuracy, and detection speed, respectively. These are quantitative comparison. (From lines 151 to 185)

Then, we showed the erroneous detection results using resultant photos and discussed the possible reasons of the erroneous detection results. In addition, we suggested some solution for the improvement of detection model. (From lines 185 to 206)

Additionally, we suggested enhanced moth detection method or model by adding unknown class images and showed enhanced detection results. (From lines 217 to 240)

We believe that those writing process for the chapter of “Results and Discussion” is suggested by editorial office and quite suitable for the understanding of readers.

Additionally, we add some discussion how the results of our research compare to results found in literature or previous studies as follows. Nonetheless, I feel sorry that I could not exactly describe quantitative comparisons between our results and those of previous studies because comparison targets such as the number of data, class, insect species, and applied methods, etc.

Comment or Suggestions 7) The Materials and Methods section is incomplete and not detailed enough. Images were collected from how many farms? How long have the traps been exposed to the insects, how long have the insects been dead (and possibly dehydrated and therefore changed shape or changed color)?

Answer or Correction 7) Moth images were taken every week after trap installed, and there was no visible changes in color or shape due to dehydration after moths were caught during 4~5 days. We revised sentences of line 114-117 to reflect your suggestion as follows.

“As shown in Figure 2, RGB images of moths in pheromone traps were weekly collected during three months period from two local farms located in the Protected Horticulture Research Institute operated by National Institute of Horticultural and Herbal Science in the Republic of Korea.”

Comment or Suggestions 8) How was the illumination adjusted for image acquisition? What illumination? What camera with what resolution was used?

Answer or Correction 8) We revised sentences of line 121-125 to reflect your suggestion as follows. “To acquire images under various conditions with a limited number of traps, images were collected using a color camera (acA2440-75uc, Basler Inc., USA) that has a 2448×2048 pixel resolution, and a 30 degree of horizontal field of view. In addition, two different adjusting illuminations (natural light, white LED light), and two photographing angles (normal directional image, oblique images) were used during the image collection.”

Comment or Suggestions 9) What was the background material?

Answer or Correction 9) Background material was plastic. We added line 117 as follows.

“The trap was made of plastic and a pheromone lure was installed in the trap,”

Comment or Suggestions 10)

In Figure 1 and 2 a white background is visible, in Figure 3 also other colors, did that not influence the results? Later in the conclusion section (L258 ff) you talk a bit about that, please consider to move this part to the material and methods section. If I look to Figure 8 c and d it seems that the other insects are also on white background only.

Answer or Correction 10)

As you commented above, all the background color for target moths collected from farms are white, but the background colors for other insects used for unknown class were quite diverse because they were collected by several methods not from real traps we used. As you are concerned, the moth detection model we developed can be influenced by the background color, however, our goal was to detect the moth which collected from our trap system, and all the background color used in our trap system were white. Therefore, it worked well, that is, the models detected the moth satisfactorily even though it has some limitation. I agree that this model has somehow limitations as you said. We are now collecting more diverse moth images from different trap systems which has different configurations, and we will develop more robust moth detection model in near future. But, I believe that you know well it takes a lot of time to get moth images from real field. In conclusion, it was possible for models to detect the target moths from the image dataset with the white color background.

Comment or Suggestions 11) It is also unclear how many extra images/object instances were created by the data augmentation. Are the numbers in table 1 before or after data augmentation?

Answer or Correction 11) The number of images in Table 1 is the number before data augmentation. Since data augmentation is performed randomly whenever the image is repeatedly trained on the network, the number of images after augmentation cannot be accurately identified.

Comment or Suggestions 12)

Also, what was the specs of the computer used for analysis (with respect to be able to asses the inference time)?

Answer or Correction 12)

We added a sentence at line 176-178 as follows.

“The training and evaluation process were carried out using a computing system with a GeForce GTX 1080ti (Nvidia Corp., Santa Clara, CA, USA) GPU and an Intel core i7-7700k (Intel Corp., Santa Clara, CA, USA) CPU.”

Comment or Suggestions 13)

A comment/question on your method: when the deep-learning network detects an object it also provides a confidence score (0 to 100%) about the detection (typically shown in the upper right corner of the bounding box). It remains unclear if you used all detections or if you applied a threshold to this score in order to count only detections with a high confidence. Please elaborate on this. Using such a threshold may also be helpful to exclude false positives for the three-class detector (as shown in figure 8 c) as most likely such detections have a low confidence value.

Answer or Correction 13)

As you commented, the threshold value is very important because it determines the precision and recall values. The results of the paper's detection images are based on a confidence score threshold of 50%. And, we revised the captions of figure 7 at line 246 and figure 8 at line 252 to reflect your comments as follows.

“Figure 7. Moth detection results of Faster R-CNN ResNet 101 detector with a confidence score threshold of 50%;”

“Figure 8. Erroneous detection results with a confidence score threshold of 50%;”

Also, we believe you know well that we can adjust intensity of the desired detection by adjusting it, and the adjustment can be determined depending on its purpose, whether to increase the Recall or increase the Precision.

Comment or Suggestions 14) More detailed comments: L33: “more stable detection” more stable than what? The detection has a certain precision and accuracy and results between different methods might be different. But stable is a very subjective word that should be avoided.

Answer or Correction 14)

As you commented, stable is an ambiguous expression. So we revised it at line 32 as follows.

“Furthermore, when unexpected insects were included in the collected images, a four-class detector with an unknown class (non-target insect) showed lower detection error than a three-class detector.”

Comment or Suggestions 15)

L42: Please elaborate why real-time monitoring is needed. I do see that automatic procedures are needed, but why real-time. You have first to define what you understand as real-time.

Answer or Correction 15)

Word “real-time” in that line is not intended to describe the full real-time, but is an expression used as the opposite of manual monitoring, where the monitoring cycle must be lengthened. As you suggested, it seems that automatic is appropriate for the line, so we revised line 43-44 as follows.

“In particular, early diagnosis of pests, one of the major challenges in the horticulture industry, requires automatic monitoring rather than manual monitoring [2].”

Comment or Suggestions 16)

L47: RGB color sensor. I guess you mean color camera, please be more specific as a color sensor does not necessarily needs to generate 2d images.

Answer or Correction 16)

We revised line 47-50 as follows.

“Among various vision systems, real-time monitoring equipment using an RGB (Red, Green, and Blue) color camera is fairly configurable because of its relatively inexpensive cost, and it can obtain useful information including shapes, colors and textures.”

Comment or Suggestions 17)

L64: “reduces the dectection speed” is wrong. It increases the detection speed (or reduces the time needed).

Answer or Correction 17)

We revised the sentence at line 81-83 as follows.

“Faster R-CNN, developed from R-CNN, increases the detection speed by applying a region proposal network (RPN), enabling real-time deep-learning-based detection.”

Comment or Suggestions 18)

L140/L141 reference 27 is mentioned before 26. Numbers should be in ascending numbers according to the appearance in the text.

Answer or Correction 18)

For a revised manuscript, the numbers of reference [26], [27] were changed to [35], [36], respectively.

Reference [26] was cited twice in this study first at line 167 and secondly at line 176, and the reference [27] was cited at line 175. Therefore, the reference [27] at line 175 was written before a reference [26] at line 176.

Comment or Suggestions 19) Table 2: I would rather call it “ms/image” instead of “ms/photograph”.

Answer or Correction 19) As you commented, we revised “ms/photograph” to “ms/image” in Table 2.

Comment or Suggestions 20) L219: minute improvement? Do you mean: minor improvement?

Answer or Correction 20)

The sentence means that an accuracy was increased even though one more class was added, but it seems that it has been written incorrectly during writing process. The sentence was revised at line 256-259 as follows.

“For comparison of average precision in the same test set, a test set was constructed with only target moth data images excluding unknown class images. Although new class (unknown class) was added on the model and only moth trap images were tested, there was improvement in mean average precision.”

Comment or Suggestions 21)

L246: you talk about “good” performance. However, it remains completely undefined what good or bad is.

Answer or Correction 21)

We revised it at line 290 as follows.

“Although we suggested deep learning-based moth detection models with high performance of mAP 90.25 through a training process with image datasets consisting of both moth images from trap and non-target insect images, higher performance will be expected if more moth images from traps and diverse environments are collected and used in the training process.”

Comment or Suggestions 22)

L270: also here you say that it “worked well”. This is very subjective and should be avoided. I miss the evaluation against the (to be defined) requirements or how your results compare to other research.

Answer or Correction 22)

We also wanted to compare the accuracy with the previous study, but we could not make a quantitative comparison because the classes and target types, etc., which could affect the quantitative accuracy comparison, were very different for each study. As your comment, “worked well” is an ambiguous expression, so we revised it at line 312 as follows.

“Despite some limitations in this study, we observed that the developed deep learning-based moth detection models showed a mAP of 90.25 for the detection of three species of pest moths in horticulture farms.”

Round 2

Reviewer 2 Report

Thank you for adressing in this revision all the comments from my review in an appropriate way.